# Evaluating a Single Domain Antibody Targeting Human PD-L1 as a Nuclear Imaging and Therapeutic Agent

**DOI:** 10.3390/cancers11060872

**Published:** 2019-06-21

**Authors:** Katrijn Broos, Quentin Lecocq, Catarina Xavier, Jessica Bridoux, Tham T. Nguyen, Jurgen Corthals, Steve Schoonooghe, Eva Lion, Geert Raes, Marleen Keyaerts, Nick Devoogdt, Karine Breckpot

**Affiliations:** 1Laboratory for Molecular and Cellular therapy (LMCT), Vrije Universiteit Brussel, Laarbeeklaan 103, B-1090 Brussels, Belgium; katrijn.broos@vub.ac.be (K.B.); quentin.lecocq@vub.be (Q.L.); jurgen.corthals@vub.be (J.C.); 2In Vivo Cellular and Molecular Imaging Laboratory (ICMI), Vrije Universiteit Brussel, Laarbeeklaan 103, B-1090 Brussels, Belgium; catarina.xavier@vub.be (C.X.); Jessica.Bridoux@vub.be (J.B.); thamnt.vn@gmail.com (T.T.N.); 3Laboratory of Cellular and Molecular Immunology (CMIM), Vrije Universiteit Brussel, Pleinlaan 2, B-1050 Brussels, Belgium; steve.schoonooghe@vub.vib.be (S.S.); geert.raes@vub.be (G.R.); 4Myeloid Cell Immunology Lab, VIB Center for Inflammation Research Center, Brussels, Pleinlaan 2, B-1050 Brussels, Belgium; 5Laboratory of Experimental Hematology, Vaccine and Infectious Disease Institute (VAXINFECTIO), Faculty of Medicine and Health Sciences, University of Antwerp, Universiteitsplein 1, B-2610 Antwerp, Belgium; eva.lion@uantwerpen.be; 6Center for Cell Therapy and Regenerative Medicine, University Hospital Antwerp, Wilrijkstraat 10, B-2650 Antwerp, Belgium; 7Department of Nuclear Medicine (NUGE), UZ Brussel, Laarbeeklaan 101, B-1090 Brussels, Belgium

**Keywords:** cancer, immune checkpoint, PD-1, PD-L1, single domain antibody, nanobody, monoclonal antibody, avelumab, immunotherapy, T cell

## Abstract

The PD-1:PD-L1 immune checkpoint axis is central in the escape of cancer cells from anticancer immune responses. Monoclonal antibodies (mAbs) specific for PD-L1 have been approved for treatment of various cancer types. Although PD-L1 blockade has proven its merit, there are still several aspects that require further attention to fully capitalize on its potential. One of these is the development of antigen-binding moieties that enable PD-L1 diagnosis and therapy. We generated human PD-L1 binding single domain antibodies (sdAbs) and selected sdAb K2, a sdAb with a high affinity for PD-L1, as a lead compound. SPECT/CT imaging in mice following intravenous injection of Technetium-99m (^99m^Tc)-labeled sdAb K2 revealed high signal-to-noise ratios, strong ability to specifically detect PD-L1 in melanoma and breast tumors, and relatively low kidney retention, which is a unique property for radiolabeled sdAbs. We further showed using surface plasmon resonance that sdAb K2 binds to the same epitope on PD-L1 as the mAb avelumab, and antagonizes PD-1:PD-L1 interactions. Different human cell-based assays corroborated the PD-1:PD-L1 blocking activity, showing enhanced T-cell receptor signaling and tumor cell killing when PD-1^POS^ T cells interacted with PD-L1^POS^ tumor cells. Taken together, we present sdAb K2, which specifically binds to human PD-L1, as a new diagnostic and therapeutic agent in cancer management.

## 1. Introduction

Cancer immunotherapy, which has the purpose to increase the pool of tumor-reactive cytotoxic T lymphocytes (CTLs), has emerged as a promising modality to treat cancer. In particular, immune checkpoint inhibitors have changed the therapeutic landscape of advanced malignancies [1].

Inhibitory immune checkpoint receptors are expressed on CTLs to attenuate their activity. This mechanism avoids overactivation of CTLs, thereby preventing collateral damage and autoimmunity [2]. However, cancer cells exploit inhibitory immune checkpoints to dampen antitumor immune responses [3]. Cancer cells of different histology can express programmed death-ligand 1 (PD-L1, CD274, B7-H1), either constitutively or in response to immune-derived signals such as interferon (IFN) [4,5]. In addition, PD-L1 can be expressed on tumor-associated cells, among other dendritic cells, macrophages, neutrophils, endothelial cells, and fibroblasts [6,7]. The net effect of the interaction between PD-L1 and its receptor, programmed death-1 (PD-1, CD279) on activated CTLs, is that CTLs become paralyzed [8].

Monoclonal antibodies (mAbs) that block the PD-1:PD-L1 axis have been successfully used to unleash the full potential of tumor-reactive CTLs in cancer patients [9,10,11,12,13,14]. However, immune checkpoint inhibiting mAbs only work in patient subsets and can become ineffective in time [15]. Since not all patients react to anti-PD-1:PD-L1 therapy, it is important to accurately predict and monitor the response to these drugs. Immunohistochemical (IHC) detection of PD-L1 has been used to discern responders from non-responders across different cancer types. However, this method only takes a snapshot, not taking into account the heterogenic and dynamic expression of PD-L1 in primary tumors or its expression in metastasis. Noninvasive molecular imaging likely circumvents these limitations [16]. Several studies have evaluated human mAbs specific for PD-L1 for this purpose [17,18,19,20,21]. Recently, a study in patients with bladder, non-small cell lung cancer, or triple negative breast cancer evaluated the zirconium-89 (^89^Zr) labeled mAb atezolizumab to detect PD-L1 expression levels in these patients. This study showed that clinical responses correlated better to PD-L1 expression levels evaluated with molecular imaging compared to IHC. However, this study also showed that maximum uptake of the radiotracer in tumor lesions was only achieved after seven days, highlighting the need to develop tracers with faster kinetics [22]. Single domain antibodies (sdAbs), which are the small antigen-binding parts of heavy-chain-only antibodies, and are otherwise known as nanobodies, have been extensively studied for imaging of tumor markers, showing that these tracers are characterized by a fast tumor uptake [23]. Previously, we generated sdAbs targeting mouse PD-L1. We showed that these radiolabeled sdAbs can be used in SPECT/CT imaging to detect heterogeneous PD-L1 expression in syngeneic mouse tumor models as fast as one hour after injection even when expression levels are low [24]. Because of their fast tumor penetrating properties, sdAbs might also represent interesting therapeutic agents.

Here, we describe a high-affinity anti-human PD-L1 sdAb, referred to as sdAb K2. This sdAb binds to the same epitope on PD-L1 as the mAb avelumab. Nuclear imaging in several cancer models and different in vitro cell-based functional assays show that sdAb K2 has high potential as a diagnostic and therapeutic agent, offering perspectives in several immune-oncology applications.

## 2. Results

### 2.1. Characterization of sdAb K2

We previously described the development of sdAbs for nuclear imaging of mouse PD-L1 in syngeneic mouse models [24]. Although affinities of the lead sdAbs for mouse PD-L1 were good, no sdAbs were found that strongly bound to human PD-L1. Therefore, in the current study, we generated a new panel of sdAbs by new alpaca immunizations, biopannings, and screenings on recombinant human PD-L1 protein, and selected among them sdAb K2 as a lead sdAb. We showed, using surface plasmon resonance (SPR), that sdAb K2 has a nanomolar affinity (K_D_ = 3.75 nM), close to the affinity of avelumab (K_D_ = 1.6 nM) for human PD-L1 (Figure 1A). However, no signal for mouse PD-L1 was detected with SPR. In addition to binding on recombinant human PD-L1 protein, sdAb K2 and avelumab were able to bind to human PD-L1 on PD-L1^POS^ 293T cells (Figure 1B).

### 2.2. sdAb K2 Generates Strong Positive Contrast in Xenograft Tumor Models Using SPECT/CT Imaging

We radiolabeled sdAb K2 and the unrelated sdAb R3B23 with ^99m^Tc through complexation of ^99m^Tc-tricarbonyl with the sdAb’s HIS-tag followed by purification by filtration and NAP5 column. This resulted in a radiochemical purity of >98% for both sdAb K2 and sdAb R3B23 as determined by Instant Thin Layer Chromatography (iTLC).

The biodistribution of sdAb K2 was first evaluated in healthy C57BL/6 mice using SPECT/CT imaging and compared to that of sdAb R3B23. As can be expected from small hydrophilic proteins with fast pharmacokinetics and lack of a molecular target in naive mice, in vivo image analysis, one hour after intravenous (i.v.) administration of both tracers, only showed radioactive signals in the kidneys and bladder, and low signals in all other organs (Figure 2A). Remarkably, kidney retention of ^99m^Tc-K2 was much lower than ^99m^Tc-R3B23. Quantification of tissue uptake levels by ex vivo γ-counting, confirmed this observation with uptake in kidneys of sdAb K2 the lowest ever reported for a ^99m^Tc-labeled sdAb under the same conditions (34.0 ± 0.82 %IA/g) (Figure 2B).

We next transplanted MCF7 breast cancer or 624-MEL melanoma cells (PD-L1^NEG^), or their PD-L1 engineered counterparts (PD-L1^POS^) in nude mice and followed tumor growth (Appendix A). SPECT/CT was performed on day 30 of tumor growth in the breast cancer model (Figure 3A), generating strong positive contrast images for PD-L1^POS^ but not PD-L1^NEG^ MCF7 tumors (Figure 3B). Ex vivo γ-counting confirmed accumulation of sdAb K2 in PD-L1^POS^ MCF7 tumors (3.07 ± 0.24 %IA/g) when compared to PD-L1^NEG^ MCF7 tumors (0.73 ± 0.16%IA/g) (Figure 3C) with high tumor-to-blood ratios in the PD-L1^POS^ tumor (Figure 3D). Flow cytometry on single-cell suspensions from these tumors confirmed expression of PD-L1 on cells obtained from PD-L1^POS^ MCF7 tumors but not PD-L1^NEG^ MCF7 tumors (Figure 3E).

A similar experiment was performed using PD-L1^POS^ and PD-L1^NEG^ 624-MEL melanoma cells (Figure 4A), showing that K2 selectively accumulates in PD-L1^POS^ 624-MEL tumors, generating strong contrast images (Figure 4B). These findings were again corroborated by ex vivo γ-counting with high tumor uptake values and high tumor-to-blood ratios in the PD-L1^POS^ tumor (Figure 4C,D). Flow cytometry on single cell suspensions from these tumors also confirmed expression of PD-L1 on PD-L1^POS^ 624-MEL cells but not on PD-L1^NEG^ 624-MEL cells (Figure 4E).

### 2.3. sdAb K2 Detects Human PD-L1 Expression in Response to IFN-γ in Xenograft Tumor Models

Following validation in two PD-L1 engineered tumor cell mouse models, we evaluated whether sdAb K2 can be used to detect PD-L1 expression in response to IFN-γ. The 938-MEL model was used as we observed in flow cytometry that in vitro treatment of 938-MEL cells with 100 IU/mL IFN-γ leads to upregulation of PD-L1 (Figure 5A). We next injected recombinant IFN-γ in 938-MEL tumors grown in athymic nude mice and used ^99m^Tc-K2 and SPECT/CT imaging to evaluate PD-L1 expression (Figure 5B). Tumors of on average 150 mm^3^ were injected with PBS (control) or 10^4^ IUs IFN-γ. One day later, we performed SPECT/CT imaging, showing detection of PD-L1 in IFN-γ but not of PBS treated tumors (Figure 5C). Furthermore, ex vivo γ-counting showed higher uptake of ^99m^Tc-K2 in mice treated with IFN-γ (0.55 ± 0.08%IA/g) compared to mice treated with PBS (0.28 ± 0.02%IA/g) (Figure 5D). Evaluation of PD-L1 expression on tumor cells using flow cytometry confirmed higher PD-L1 expression on IFN-γ treated tumors compared to PBS treated tumors, although PD-L1 expression levels were low (Figure 5E).

### 2.4. sdAb K2 Competes with Avelumab for Binding to PD-L1 and Has PD-1:PD-L1 Blocking Capacity

We showed that sdAb K2 serves as a potential diagnostic agent to detect PD-L1 expression levels in vivo on tumor cells and as such might select patients for anti-PD-L1 treatment. We next wondered whether sdAb K2 also has therapeutic potential and evaluated whether sdAb K2 is able to inhibit the PD-1:PD-L1 interaction leading to enhanced T-cell activity. We showed by SPR that sdAb K2 competes with avelumab for binding to PD-L1, since relative response units, or in other words binding to immobilized human PD-L1, is reduced when avelumab is added to sdAb K2 and vice versa (Figure 6A). Moreover, we showed using SPR that sdAb K2 is able to inhibit the interaction between PD-1:PD-L1 by half with a concentration of 8.5 nM (IC_50_). In the same assay, the IC_50_ value of avelumab was 4 nM, whereas both controls, sdAb R3B23, and trastuzumab, did not influence the PD-1:PD-L1 interaction (Figure 6B).

### 2.5. sdAb K2 Facilitates T-Cell Receptor (TCR) Signaling in 2D3 Reporter Cells upon Recognition of Tumor Cells

We furthermore used a cell-based assay to evaluate whether sdAb K2 is able to block the PD-1:PD-L1 pathway leading to enhanced TCR signaling. This assay uses 2D3 reporter cells that are derived from Jurkat T cells. The 2D3 reporter cells express CD8 and lack an endogenous TCR. We engineered them to express both the TCRαβ recognizing gp100_280–288_ in the context of HLA-A2 and PD-1 (PD-1^POS^ 2D3). When the TCR is triggered, the cells express enhanced green fluorescent protein (eGFP) under the control of the nuclear factor of activated T cells (NFAT) promoter [25].

We used this platform to evaluate the blocking ability of sdAb K2, as schematized in Figure 6C,D. In particular, HLA-A2^POS^ PD-L1^POS^ or PD-L1^NEG^ 624-MEL melanoma or MCF7 breast cancer cells (Figure 6C) were pulsed with the gp100_280–288_ peptide. These cells were co-cultured with PD-1^POS^ 2D3 cells expressing the TCRαβ recognizing gp100_280–288_ in the context of HLA-A2 (Figure 6D). When the gp100_280–288_ peptide is recognized by the specific TCR on the 2D3 cells, this will lead to eGFP upregulation by the 2D3 cells (Figure 6D). Co-culturing of PD-1^POS^ 2D3 cells with PD-L1^POS^ tumor cells reduced eGFP expression, a measurement for TCR triggering, compared to co-culturing PD-1^POS^ 2D3 cells with PD-L1^NEG^ tumor cells (Figure 6E,F, condition ‘no’; 0.53 ± 0.21 for the 624-MEL model and 0.71 ± 0.09 in the MCF7 model). This reflects the effect of the interaction between PD-1 and PD-L1, leading to inhibition of TCR signaling [26]. Adding 360 nM of sdAb K2 or avelumab could increase eGFP expression levels, or in other words alleviate the inhibition on TCR signaling by the PD-1:PD-L1 interaction (0.93 ± 0.05 for sdAb K2 and 0.98 ± 0.02 for avelumab). Moreover, eGFP expression of PD-1^POS^ 2D3 cells co-cultured with PD-L1^POS^ MCF7 tumor cells treated with sdAb K2 or avelumab was even higher compared to co-culturing with PD-L1^NEG^ MCF7 tumor cells (1.23 ± 0.33 for sdAb K2 and 1.28 ± 0.46 for avelumab) (Figure 6E,F).

### 2.6. sdAb K2 restores the Tumor Cell Killing Ability of Activated Peripheral Blood Mononuclear Cells (PBMCs)

We also explored whether adding sdAb K2 to co-cultures of activated PBMCs and tumor cells would result in enhanced tumor cell killing. Because CD8^POS^ T cells are critical to mediate tumor cell killing, we first evaluated the expression of PD-1 and PD-L1 on CD8^POS^ T cells, present within the pool of PBMCs, stimulated with a cocktail of anti-CD3 antibodies and IL-2. We showed, using flow cytometry, that both PD-1 and PD-L1 were upregulated on CD8^POS^ T cells, thereby confirming activation of these cells (Figure 7A). The PBMCs, containing activated CD8^POS^ T cells, were added to PD-L1^POS^ or PD-L1^NEG^ 624-MEL tumor cells that were lentivirally engineered to express eGFP and that were grown in 3D spheroids (Figure 7B). The eGFP signal can be monitored in real-time using the IncuCyte Zoom^®^ live cell imaging system and is used to measure tumor cell growth. In the absence of PD-1:PD-L1 blocking moieties, we observed that the green objective area of the eGFP^POS^ PD-L1^POS^ tumor cells increased in time, signifying that tumor cells were not killed by activated T cells, and as such, confirming the inhibitory role of PD-L1 on tumor cell killing by PD-1^POS^ CD8^POS^ T cells (Figure 7C). Addition of 3.6 μM avelumab or sdAb K2 to PD-L1^POS^ tumor cells and stimulated PBMCs reduced the green objective area, signifying that tumor cell killing occurred (green curve) when compared to the addition of a control mAbs or sdAb R3B23 (red curve) (Figure 7D,E). The effect of adding avelumab on tumor cell killing could be observed at 80 hours, which is reflected by the reduction in green objective area at that time point (Figure 7D), while the effect of sdAb K2 was observed in the first hours of culture (Figure 7E). Hence, sdAb K2 showed early and short therapeutic activity, while avelumab showed a more durable blocking activity. We next evaluated whether adding both sdAb K2 and avelumab to PD-L1^POS^ tumor cells and PD-1^POS^ stimulated PBMCs could enhance the tumor cell killing. We showed more efficient tumor cell killing (after 70 h) compared to the addition of avelumab or sdAb K2 separately (Figure 7F). Remedying the short action of sdAb K2, we also tested the effect of repeated administrations of sdAb K2 with 24 h intervals to PD-L1^POS^ tumor cells and PD-1^POS^ stimulated PBMCs. This resulted in efficient tumor cell killing by activated PBMCs after 50 hours (Figure 7G), supporting the therapeutic potential of sdAb K2.

## 3. Discussion

In this study, we showed that the sdAb, designated as sdAb K2, is able to detect human PD-L1 expression levels in the tumor environment and to block PD-L1 on tumor cells resulting in enhanced T-cell activity. sdAb K2 binds with nanomolar affinity to human PD-L1 and can be used as a diagnostic to detect PD-L1 expression in the tumor as fast as one hour after injection. PD-L1 expression could even be detected after intratumoral administration of IFN-γ, which led to in situ upregulation of PD-L1, although expression levels remained low. Moreover, we showed that sdAb K2 has therapeutic potential as it exhibits an IC_50_ of 8.5 nM to block PD-1:PD-L1 interactions and releases the break on antigen-specific TCR signaling and on tumor killing activity in vitro.

Nowadays, in clinical trials, PD-L1 expression is mainly evaluated by IHC, which has some limitations. Staining of fixed selected tissue samples does not allow assessment of heterogenic expression of tumor markers or the dynamic PD-L1 expression during treatment. Molecular imaging is a good alternative to assess PD-L1 expression, as this noninvasive method can show regional differences within the tumor environment and can assess PD-L1 expression in metastatic lesions. Here, we showed that sdAb K2 has several properties to make it an interesting diagnostic. Ideal radiotracers combine fast renal clearance and efficient tumor penetration with a good affinity for their target resulting in high tumor-to-background ratios shortly after tracer administration. We showed that ^99m^Tc-K2 fulfills these requirements since administration in healthy C57BL/6 mice revealed little to no signals in all organs except in the kidneys and urinary bladder, which is due to the renal uptake and elimination because of their small size [27]. Noteworthy, the uptake of ^99m^Tc-K2 in the kidneys was much lower compared to ^99m^Tc-R3B23, the sdAb used as a negative control, and to our knowledge any other sdAb that was labeled in a similar fashion. This low kidney retention makes sdAb K2 particularly suited as a radiotracer, since such important decrease in kidney retention not only lowers the irradiation burden for the patient but also improves the assessment of lesions in the vicinity of the kidneys. This can be useful to assess patients with renal cell carcinoma for expression of PD-L1, as these patients can derive benefit from such treatments [15]. ^99m^Tc-K2 showed intense and specific uptake in two human PD-L1-expressing tumor models, melanoma and breast cancer, with tumor-to-blood ratios of 20.2 and 8.9, respectively. Moreover, PD-L1 expression could be detected after intratumoral injection of IFN-γ, leading to elevated, albeit still low, PD-L1 expression levels on tumor cells, as confirmed with flow cytometry. In all imaging studies, high tumor-to-background uptake levels could be obtained as fast as one hour after injection. When translated to patients, this would allow short, same day imaging procedures, very similar to the current daily practice with ^18^F-FDG [28]. We observed a similar absolute tumor uptake with sdAb K2 compared to other studies using sdAbs that target tumors [29,30]. Although absolute tumor uptake for sdAbs is generally lower than what can be obtained with mAbs, the contrast that can be obtained at early time points is much higher, due to the very fast clearance of the unbound tracer. For future clinical translation, the here proposed SPECT tracer will be further engineered into a clinical PET tracer, similar to what was done for other sdAb translations [29,30,31].

Other research groups have also developed radiotracers for PD-L1 imaging using mAbs [18,22] or smaller proteins [32,33,34] of which some have entered clinical testing. Bensch et al. used ^89^Zr-labeled atezolizumab, a clinically approved therapeutic mAb, for molecular imaging in cancer patients. A better correlation between PET images and clinical responses compared to IHC was reported. However, optimal tumor-to-blood ratios were only obtained on day 7 after injection [22]. This time point could be tangibly reduced to five days using the ^89^Zr-labeled heavy chain-only antibody KN035, an anti-PD-L1 sdAb fused to an Fc domain, which is smaller (80 kDa) than a full antibody (150 kDa), however still substantially larger than sdAb K2 (15 kDa). The tumor-to-blood ratios reported with KN035 were however low, i.e., 1.1 [35]. Another compound, ^18^F-labeled adnectin (^18^F-BMS-986192), has a size of about 10 kDa, and is, therefore, at least in terms of size, closer to sdAbs. This compound could visualize PD-L1^POS^ tumors with a 3.5-fold higher uptake in PD-L1^POS^ versus PD-L1^NEG^ tumors using PET imaging in mice. Kidney uptake of ^18^F-BMS-986192 was relatively high [33]. Nonetheless, this compound was recently evaluated in non-small cell lung cancer patients. In many cases, tracer uptake in the tumor correlated with PD-L1 expression levels on tumor cells as determined by IHC. However, a subset of tumors showed low PD-L1 expression by IHC, while relatively high uptake with ^18^F-BMS-986192 was observed. This could be explained by the heterogeneity of PD-L1 in the lesion. Furthermore, response rates correlated with tracer uptake, with responders showing higher tracer uptake compared to non-responders [34]. These observations make us confident that small imaging agents, such as the here presented sdAb K2, can be used as a diagnostic tool in cancer patients. Indeed, sdAb K2 is able to image PD-L1 with high contrast levels as fast as one hour after injection, which is much faster than imaging with ^89^Zr-labeled atezolizumab. Secondly, because of its small size sdAb K2 is able to efficiently penetrate tumors resulting in higher tumor-to-blood ratios compared to compound KN035 (8.9 and 20.2 for sdAb K2 compared to 1.1 for KN035). Finally, sdAb K2 is able to detect PD-L1 expression levels with higher contrast compared to similar-sized ^18^F-BMS-986192 (4.2 and 5.7-fold higher uptake for sdAb K2 in PD-L1^POS^ compared to PD-L1^NEG^ breast or melanoma tumors respectively) and with lower kidney retention.

Besides its diagnostic value, we further evaluated the therapeutic value of sdAb K2. The use of sdAbs for therapy exhibits some advantages compared to mAbs. sdAbs are 10 times smaller than mAbs and are, therefore, better suited for fast and homogenous tumor penetration. Because the PD-1:PD-L1 immune checkpoint is highly relevant in the tumor environment, this could be a key characteristic for optimal therapeutic effect in larger, difficult-to-penetrate tumors [36]. This was already exemplified by treatment of tumors with a smaller PD-L1 blocking moiety, the HAC-I variant (10 kDa, K_D_ = 100 pM, IC_50_ = 210 pM). Treatment of small, human PD-L1^POS^ tumors with the HAC-I variant induced equal tumor reduction compared to treatment with mAbs. However, when larger tumors were treated it appeared that mAbs lost their therapeutic efficiency whereas the HAC-I variant did not [37]. Additionally, KN035 (anti-PD-L1 sdAb fused to an Fc domain) enhanced tumor cell killing in a xenograft model [38]. However, sdAbs that target human PD-L1 have not yet been studied in their monovalent format in a therapy setting. We were able to show that sdAb K2 binds to the same epitope on PD-L1 as the FDA-approved mAb avelumab and is able to block the PD-1:PD-L1 interaction in a similar magnitude as avelumab in a human antigen-specific T-cell assay, even though the IC_50_ value of sdAb K2 was slightly higher. This difference in IC_50_ can be explained by the bivalent format of avelumab, which renders two binding places for avelumab compared to one for sdAb K2. When evaluating both compounds in a 3D tumor cell killing assay, we observed that tumor cell killing occurred rapidly after addition of sdAb K2, whereas for avelumab the effect was only observed after 80 hours. This may be explained by differences in valency, IC_50_ as well as diffusion between both agents in the 3D spheroids. sdAb K2 is small and should be able to rapidly bind to its target. However, it is likely that sdAb K2 also rapidly detaches from its target. In contrast, avelumab is larger and probably reaches its target later. However, the higher avidity, due to its bivalency, is likely to result in better off-rates and longer retention times. Hence, as a therapeutic, repeated administration of sdAb K2 could be necessary to obtain the same effect as avelumab. We already confirmed in vitro that repeated administration of sdAb K2 to cultures of activated PBMCs and PD-L1^POS^ tumor cells facilitated tumor cell killing comparable to what was observed with avelumab. However, it remains to be shown if this could also improve clinical outcome. Exploiting their differences in pharmacokinetics and avidity we further demonstrate that combinatorial treatment with sdAb K2 and avelumab results in a superior tumor-killing effect. Further research to determine the exact value of such a combination approach in an in vivo tumor setting is warranted. Taken together, these data show that sdAb K2 holds promise as a small antagonistic therapeutic compound targeting human PD-L1. This is further substantiated by the low immunogenicity of sdAbs [29,30,31]. Moreover, as sdAb K2 and avelumab compete for binding to human PD-L1, and as sdAbs in comparison to mAbs are faster cleared from the blood, we expect that the toxicity profile of sdAb K2 will not exceed that of avelumab. For the latter, it has been described that the treatment-related adverse events are manageable [39].

In conclusion, we described sdAb K2, a highly specific and high-affinity anti-human PD-L1 sdAb, which shows high potential as a PET tracer to diagnose PD-L1 expression in the tumor early after administration. Therapeutically, we showed that sdAb K2 binds on the same epitope on human PD-L1 as avelumab, an FDA-approved mAb developed by Merck KGaA (EMD Serono) and Pfizer and can exhibit significant antitumor activity, warranting further exploitation of sdAb K2 in immune-oncology applications.

## 4. Materials and Methods 

### 4.1. Reagents

All Biacore consumables were from GE Healthcare (Machelen, Belgium). HIS-tagged PD-L1 recombinant proteins (SINO Biologicals, Wayne, PA, USA), and recombinant Fc-tagged PD-L1 and PD-1 proteins (both R&D Systems, Minneapolis, MN, USA) were used in SPR to evaluate the affinity and IC_50_, respectively, of PD-L1 binding sdAbs and avelumab (Bavencio^®^, provided by Merck KGaA [EMD Serono], Overijse, Belgium) and Pfizer, Puurs, Belgium). A sdAb specific for a multiple myeloma paraprotein, designated sdAb R3B23 [40], and trastuzumab (Herceptin^®^, Roche, Bazel, Switzerland) served as negative controls.

Anti-HIS mAbs (Bio-Rad, Temse, Belgium, AD1.1.10) and phycoerythrin (PE)-conjugated anti-mouse IgG antibodies (BD Biosciences, Erembodegem, Belgium, A85-1) were used to detect binding of HIS-tagged sdAbs to PD-L1^POS^ 293T cells. An anti-human PE-labeled IgG1 antibody (Miltenyi Biotec, Bergisch Gladbach, Germany, IS11-12E4.23.20) was used to detect binding of avelumab to PD-L1^POS^ 293T cells. Expression of PD-L1 on cells was evaluated with anti-PD-L1 antibodies coupled to allophycocyanin (APC, eBioscience, Brussels, Belgium, MIH1) or PE-CF594 (Biolegend, San Diego, CA, USA, MIH1), HLA-A2 using a PE-conjugated anti-HLA-A2 antibody (BD Biosciences, BB7.2), PD-1 using a PE-conjugated anti-PD-1 antibody (Biolegend, EH12.2H7). 2D3 cells were discriminated from tumor cells in the 2D3 functional assay using an APC-H7-labeled anti-CD8 antibody (BD Biosciences, SK1). Expression of the TCR on electroporated 2D3 cells was evaluated with a PE-labeled anti-TCRα/β antibody (Biolegend, IP26). Isotype-matched antibodies served as controls (BD Biosciences).

The gp100_280–288_ peptide (Eurogentec, Cologne, Germany; YLEPGPVTA) was loaded onto tumor cells in the 2D3 assay. Avelumab (Bavencio^®^, provided by Merck KGaA [EMD Serono] and Pfizer), an isotype-matched control antibody (Bioxcell, West Lebanon, NH, USA, MOPC-21) and R3B23 were used in the 2D3 and 3D spheroid assays as controls.

### 4.2. Generation, Selection, Production and Purification of sdAbs that Bind Human PD-L1

Alpacas were immunized subcutaneously for six times at a weekly interval with 100 μg recombinant PD-L1-Fc protein (R&D Systems). Peripheral blood lymphocytes were purified and used as a source to create a sdAb phage display library in the pMECS plasmid. This library was used for biopanning on the immunogen, and periplasmatic extracts of individual sdAb clones were tested in ELISA [41,42]. Sequence analysis was performed on sdAb clones that specifically bound PD-L1.

Anti-PD-L1 sdAbs and the control sdAb R3B23 were produced and purified as described in [24]. Therefore, the sdAb cDNA was cloned in the vector pHEN6 to incorporate a C-terminal HIS-tag.

### 4.3. Surface Plasmon Resonance

The affinity of sdAbs was evaluated on immobilized PD-L1 proteins, as described in [24]. To evaluate the sdAb’s IC_50_, Fc-PD-1 protein was immobilized on a CM5 chip. Varying sdAb concentrations (400 to 0.78 nM using a two-fold dilution series) were mixed with 25 nM recombinant Fc-PD-L1 protein and run over the chip. The maximum relative response values were plotted in function of competing sdAb concentration in Prism to calculate IC_50_ values. To evaluate competition between sdAbs and avelumab for binding to PD-L1, competition studies were performed as described in [28].

### 4.4. Mice and Cell Lines

Female, six-week-old C57BL/6 and Crl:NU(NCr)-Foxn1nu (nude) mice were supplied by Charles River (Ecully, France). Human 293T cells and HLA-A*0201^POS^ breast carcinoma cells (MCF7) were purchased from the American Type Culture Collection (ATCC) and cultured according to ATCC’s recommendation. HLA-A*0201^POS^ 624-MEL or 938-MEL cells were provided by S.L. Topalian (National Cancer Institute, Baltimore, MD, USA). 624-MEL and 938-MEL cells were cultured in RPMI1640 medium supplemented with 10% Fetal clone I serum (Thermoscientific, Brussels, Belgium), 2 mM L-Glutamine, 100 U/mL penicillin, 100 µg/mL streptomycin, 1 mM sodium pyruvate and nonessential amino acids (Sigma-Aldrich, Ghent, Belgium). PD-1^POS^ 2D3 cells were generated and cultured, as described in [25].

Experiments were performed using blood samples from healthy HLA-A*0201^POS^ donors provided by the Blood Transfusion Center of the Brussels University Hospital. Isolation of PBMCs was performed, as described in [43].

### 4.5. Lentiviral Vector Production, Characterization and Transduction

The plasmids pCMVΔR8.9 and pMD.G were a gift from D. Trono (Ecole Polytechnique Fédéral de Lausanne, Lausanne, Switzerland). The transfer plasmids encoding eGFP, PD-L1 and PD-1 were described in [44,45]. Lentiviral vectors were produced and characterized, as described in [46]. 293T, MCF7, and 624-MEL cells were transduced with PD-L1 or eGFP-encoding lentiviral vectors at a multiplicity of infection (MOI) of 10, while 2D3 cells were transduced with PD-1 encoding lentiviral vectors at an MOI of 5.

### 4.6. Tumor Challenge and Preparation of Tumor Single Cell Suspensions

Athymic nude mice were injected s.c. with 5 × 10^6^ MCF7, 624-MEL, 938-MEL or PD-L1-modified MCF7 or 624-MEL cells. One day before transplanting MCF7 cells, mice were implanted with estrogen pellets (Innovative research of America; 0.36 mg/mouse). The tumor length and width were measured using an electronic caliper and used to calculate the tumor volume using the formula: (length × width^2^)/2. One day prior to imaging, 938-MEL tumor-bearing mice were injected intratumorally with 50 µL phosphate buffered saline (PBS; Sigma-Aldrich) or IFN-γ (2 × 10^6^ IUs/mL, ImmunoTools). Tumor tissue was reduced to single cells using the GentleMACS tumor dissociation protocol (Miltenyi Biotec) [47].

### 4.7. Pinhole SPECT-Micro-CT Imaging and Image Analysis

Labeling of sdAbs with ^99m^Tc was performed, as described in [48]. One hour prior to imaging, mice were injected intravenously with 100–200 µL of 45–155 MBq of ^99m^Tc-labeled sdAbs (10 µg). Pinhole SPECT-micro-CT imaging and image analysis in naive C57BL/6 mice, the MCF7 and 624-MEL tumor model were performed, as described in [24]. For the 938-MEL model, SPECT/CT was performed on a MILabs VECTor/CT camera. The CT-scan was set to 60 kV and 615 mA. CT scan time was 139 seconds. SPECT-images were obtained using a rat SPECT-collimator (1.5-mm pinholes) in spiral mode, six positions for whole-body imaging, with 150 seconds per position, total body SPECT scan was 15 min. Images were reconstructed with 0.4 mm voxels with two subsets and four iterations, without a post-reconstruction filter. Images were further visually analyzed and quantified where appropriate, using AMIDE (Medical Image Data Examiner software, http://amide.sourceforge.net/). Maximum intensity projections (MIP) were generated using OsiriX Lite software. After imaging, mice were sacrificed and selected organs were isolated to measure radioactivity using a γ-counter (Cobra Inspector 5003, Packard, Machelen, Belgium). The amount of radioactivity in organs is expressed as percent injected activity per gram (%IA/g).

### 4.8. mRNA Production and Electroporation of Cells

The pGEM vectors encoding the human gp100 TCRα and TCRβ were kindly provided by Prof. N. Schaft (Universitätsklinikum Erlangen, Erlangen, Germany) [25]. The production, purification, quantification and quality control of mRNA was performed, as described in [43]. Human gp100 TCRα and TCRβ mRNA (2.5 µg each/10E6 cells) was electroporated into PD-1^POS^ 2D3 cells in 200 µL OptiMEM medium (Life Technologies, Erembodegem, Belgium) in a 4 mm cuvette (Cell Projects, Kent, UK) using a time constant protocol (300 V, 7 ms) and the Gene Pulser XcellTM device (BIORAD).

### 4.9. PD-1^POS^ 2D3 Assay

The PD-1^POS^ 2D3 assay was performed as previously described [25]. Briefly, PD-1^POS^ 2D3 cells, electroporated to express the TCRαβ recognizing the gp100_280–288_ peptide (YLEPGPVTA) restricted to HLA-A2, were plated in a 96-well round-bottom plate at 10^5^ cells in 100 µL IMDM containing 10% FBS (triplicate). MCF7, 624-MEL or PD-L1 engineered MCF7 or 624-MEL cells were pulsed with 50 µg/mL gp100_280–288_ peptide and added to the cultures at effector-to-stimulator ratios of 10-to-1 in 50 µL medium. Co-cultures were performed for 24 h at 37 °C, 5% CO_2_ in the presence of 360 nM avelumab or K2. Isotype-matched mAbs or R3B23 were used as controls. The activation of PD-1^POS^ 2D3 cells was measured in flow cytometry as the percentage of eGFP^POS^ cells within CD8^POS^ 2D3 cells.

### 4.10. 3D Spheroid Cytotoxicity Assay

624-MEL cells engineered to express eGFP and PD-L1, were plated at 200 cells in an ultralow attachment 96-well plate (Costar^®^, ref 7007) and kept in culture for one day to form 3D spheroids. PBMCs stimulated for 24 h with 10 ng/mL IL-2 (Peprotech) and 10 ng/mL anti-CD3 mAbs (BioLegend, ref. 317302) were added to the cells at a ratio of 1:50 in the presence of 3.6 µM avelumab, isotype-matched mAbs, sdAb K2, sdAb R3B23, or the combination of mAbs and sdAbs. In a separate assay, sdAb K2 or sdAb R3B23 were added every 24 h to the co-culture after centrifuging the plate at 1200 rpm for 10 min and replacing 50 µL of the co-culture by 50 µL containing 3.6 µM sdAb K2 or sdAb R3B23. The reduction of the total amount of green object area within each well containing eGFP^POS^ and PD-L1^POS^ cells was evaluated every hour for seven consecutive days in an IncuCyte Zoom^®^ live cell imaging system (EssenBio, Welwyn Garden City, UK).

### 4.11. Flow Cytometry

The procedure for staining of cell surface markers was described in [45]. All cells were acquired on an LSRFortessa flow cytometer (BD Biosciences) and data were analyzed with FACSDiva (BD Biosciences) or FlowJo (Tristar Inc., Phoenix, AZ, USA) software.

### 4.12. Statistical Analysis

Results are expressed as mean ± standard error of the mean (SEM). A one-way Anova followed by Bonferroni correction or a two-tailed t-test was carried out to compare data sets. Statistical method, sample size and number of times experiments were repeated, are indicated in the figure legends.

### 4.13. Ethics Approval and Consent to Participate

Experiments were performed using blood samples from anonymous, healthy HLA-A*0201^POS^ donors provided by the Blood Transfusion Center of the University Hospital Brussels. The Ethics Committee of the University Hospital Brussels approved the study (license 2013/198). Experiments with mice were performed according to the European guidelines for animal experimentation under licenses LA1230214 and LA1230272. The Ethical Committee for use of laboratory animals of the Vrije Universiteit Brussel approved the experiments (project 15-214-1 and 18-272-12).

### 4.14. Data and Materials Availability

All data generated during this study are included in this published article.

## 5. Conclusions

The high affinity anti-human PD-L1 specific single domain antibody, designated K2, is a new diagnostic and therapeutic agent for the management of cancer.

## 6. Patents

Broos K.; Lecocq Q.; Raes G.; Bridoux J.; Devoogdt N.; Keyaerts M.; Breckpot K. have patents on the use of K2 for imaging and therapy purposes (Broos, K.; Van Ginderachter, J.; et al. human PD-L1 binding immunoglobulins: EP 18159388.0. / JoVG/huPDL1/626, 2018).

## Figures and Tables

**Figure 1 cancers-11-00872-f001:**
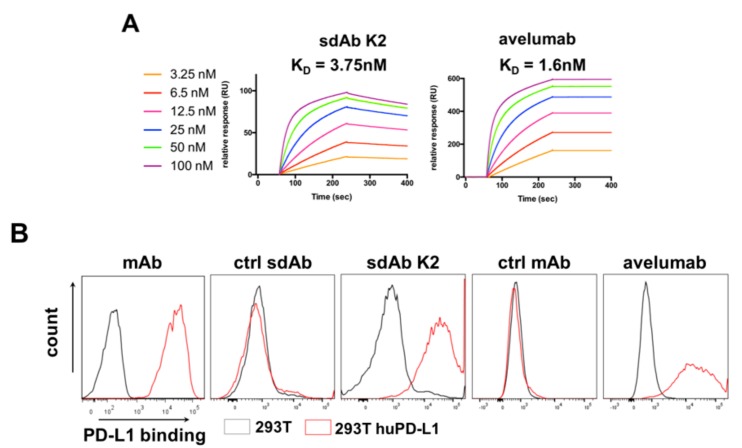
Characterization of sdAb K2, a sdAb that specifically binds to human PD-L1. (**A**) Representative sensograms showing the affinity/kinetics of different concentrations of purified sdAb K2 or avelumab, interacting with immobilized recombinant human PD-L1 protein, as determined in surface plasmon resonance (SPR). (**B**) Representative flow cytometry histograms showing labeling of PD-L1^NEG^ (grey) versus PD-L1^POS^ (red) 293T cells with mAbs specific for PD-L1, ctrl sdAb (sdAb R3B23), sdAb K2, control mAb (isotype-matched) or avelumab.

**Figure 2 cancers-11-00872-f002:**
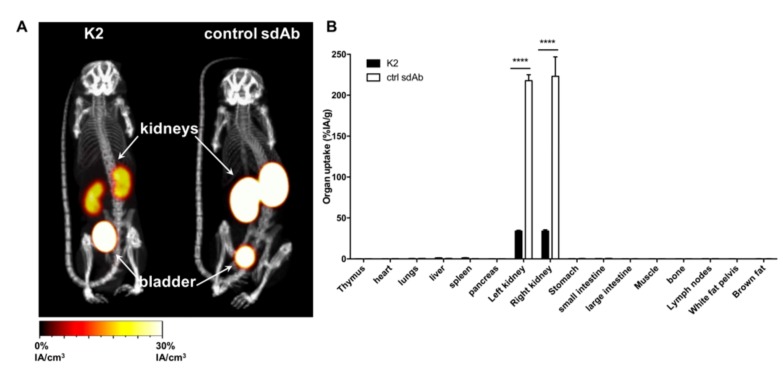
Radiolabeled sdAb K2 shows little signal in healthy mice. (**A**) SPECT/CT images showing the biodistribution of ^99m^Tc-K2 or ^99m^Tc-R3B23 (control sdAb) 1 h after i.v. administration in healthy C57BL/6 mice (*n* = 3). (**B**) Ex vivo analysis of the biodistribution of ^99m^Tc-K2 or ^99m^Tc-R3B23 (control sdAb) in dissected tissues and organs 80 minutes after i.v. administration in healthy C57BL/6 mice (expressed as percent injected activity per gram, %IA/g; n = 3). **** *p* < 0.0001.

**Figure 3 cancers-11-00872-f003:**
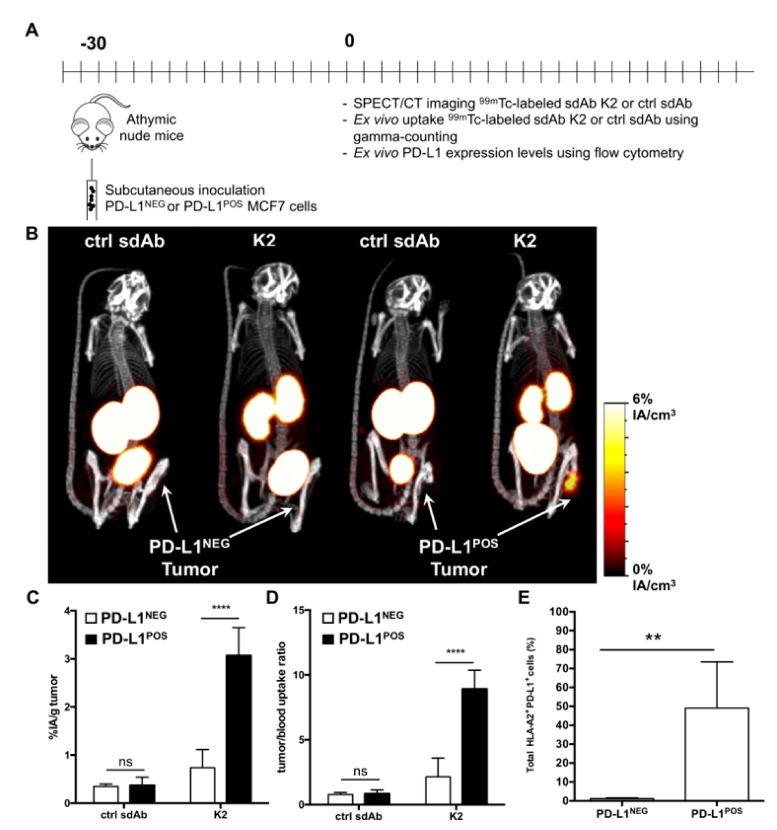
Radiolabeled sdAb K2 allows visualization of human PD-L1^POS^ breast tumors by nuclear imaging. (**A**) Scheme of the experimental setup. (**B**) SPECT/CT images showing the biodistribution of ^99m^Tc-K2 or ^99m^Tc-R3B23 (control sdAb) one hour after i.v. administration in athymic nude mice bearing PD-L1^NEG^ (**left**) or PD-L1^POS^ (**right**) MCF7 tumors (*n* = 6). (**C**,**D**) Ex vivo analysis of the accumulation of ^99m^Tc-sdAbs in dissected PD-L1^NEG^ or PD-L1^POS^ MCF7 tumors (**C**, expressed as %IA/g), and of tumor-to-blood uptake ratios (**D**), 80 min after i.v. radiotracer injection (*n* = 6). (**E**) Percentage of human PD-L1^POS^ HLA-A2^POS^ cells in tumors dissected from mice that were s.c. implanted with parental MCF7 cells (PD-L1^NEG^) or PD-L1-transduced counterparts (PD-L1^POS^), as measured by flow cytometry analysis of tumor single cell suspensions (*n* = 6). ** *p* < 0.01, **** *p* < 0.0001.

**Figure 4 cancers-11-00872-f004:**
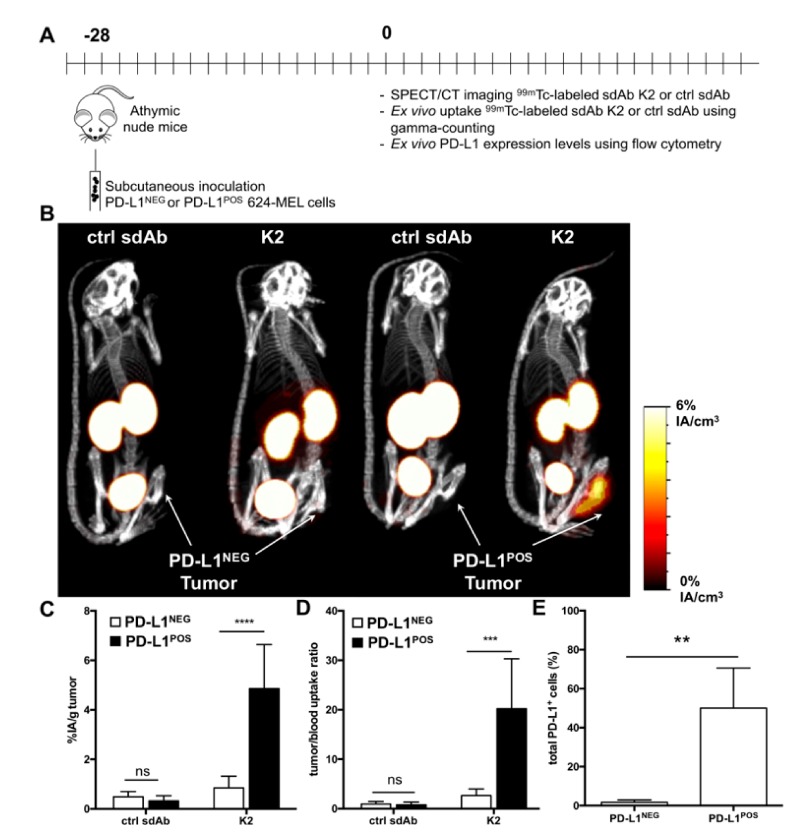
Radiolabeled sdAb K2 allows visualization of human PD-L1^POS^ melanoma tumors by nuclear imaging. (**A**) Scheme of the experimental setup. (**B**) SPECT/CT images showing the biodistribution of ^99m^Tc-K2 or ^99m^Tc-R3B23 (control sdAb) one hour after i.v. administration in athymic nude mice bearing PD-L1^NEG^ (**left**) or PD-L1^POS^ (**right**) 624-MEL tumors (*n* = 6). (**C**,**D**) Ex vivo analysis of the accumulation of ^99m^Tc-sdAbs in dissected PD-L1^NEG^ or PD-L1^POS^ 624-MEL tumors (**C**, expressed as %IA/g), and of tumor-to-blood uptake ratios (**D**), 80 min after i.v. radiotracer injection (*n* = 6). (**E**) Percentage of human PD-L1^POS^ cells in tumors dissected from mice that were s.c. implanted with parental 624-MEL cells (PD-L1^NEG^) or PD-L1-modified counterparts (PD-L1^POS^), as measured by flow cytometry analysis of tumor single cell suspensions (*n* = 6). ** *p* < 0.01, *** *p* < 0.001, **** *p* < 0.0001.

**Figure 5 cancers-11-00872-f005:**
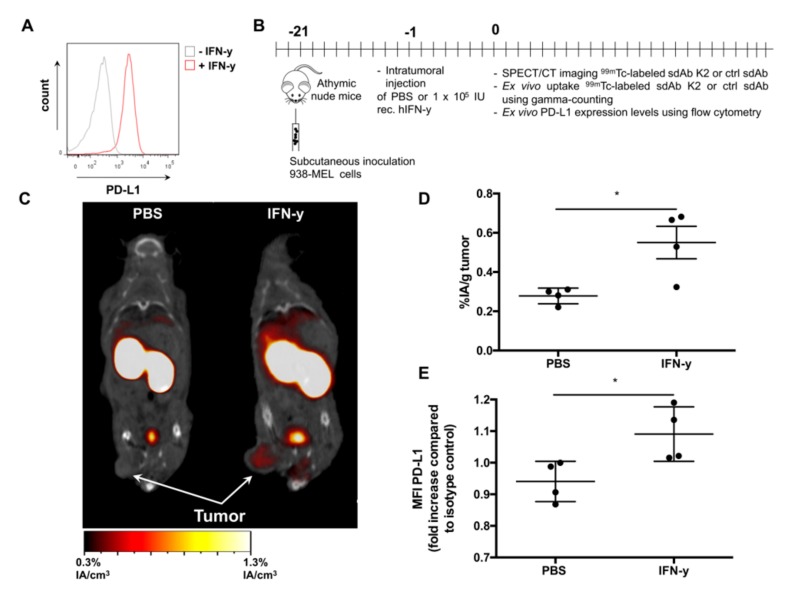
Radiolabeled sdAb K2 allows specific visualization of human PD-L1 induced by IFN-γ in 938-MEL tumors. (**A**) Representative histogram showing PD-L1 expression after in vitro stimulation of 938-MEL cells with 100 IU/mL recombinant human IFN-γ (red) or non-treated 938-MEL cells (grey), as evaluated by flow cytometry analysis. (**B**) Scheme of the experimental setup. (**C**) Representative SPECT/CT images one hour after i.v. administration of ^99m^Tc-K2 in athymic nude mice bearing PD-L1^NEG^ 938-MEL tumors that were injected i.t. with PBS (**left**) or human recombinant IFN-γ (**right**) (*n* = 4). (**D**) Accumulation of ^99m^Tc-K2 in 938-MEL tumors (expressed as %IA/g) that were treated i.t. with PBS or IFN-γ, as determined by γ-counting of dissected tumors 80 minutes after radiotracer injection (*n* = 4). (**E**) Human PD-L1 expression levels in 938-MEL tumors treated i.t. with PBS or IFN-γ. PD-L1 expression was evaluated by flow cytometry analysis of dissected and dissociated tumors. PD-L1 levels were depicted as fold increase in mean fluorescence intensity (MFI) as compared to isotype control staining (*n* = 4). * *p* < 0.05.

**Figure 6 cancers-11-00872-f006:**
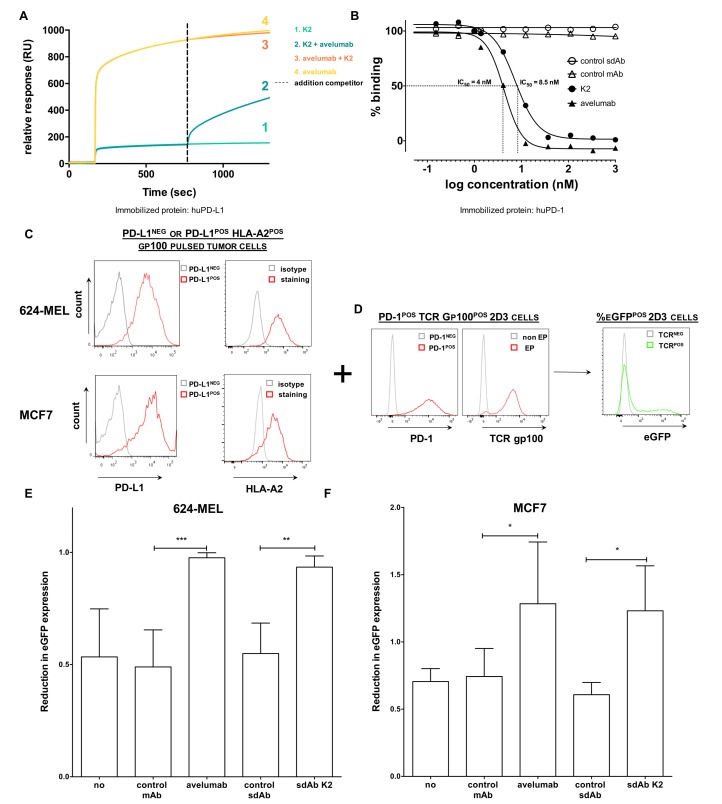
sdAb K2 and avelumab antagonize PD-1:PD-L1 interactions. (**A**) A competition study with equimolar amounts of avelumab and sdAb K2 shows that both bind the same epitope on human PD-L1 as determined by SPR. (**B**) Dose response curves of soluble recombinant human PD-L1 protein mixed with increasing concentrations of sdAb K2, control sdAb, avelumab or control mAb, to immobilized human PD-1 recombinant protein, as determined by SPR. Per condition, maximal signal is shown relative to the sample that only contains recombinant PD-L1 protein. A representative experiment out of two independent experiments is shown. (**C**,**D**) Schematic representation of the experimental set up of the PD-1^POS^ 2D3 reporter assay. (**C**) Representative flow cytometry histograms showing PD-L1 and HLA-A2 expression in HLA-A2^POS^ PD-L1^NEG^ (grey) or HLA-A2^POS^ PD-L1^POS^ (red) 624-MEL or MCF7 cells. (**D**) Representative flow cytometry histograms showing PD-1^NEG^ (grey) or PD-1^POS^ (red) 2D3 cells electroporated with mRNA encoding a TCR recognizing gp100 in the context of HLA-A2 (red). HLA-A2^POS^ PD-L1^POS^ or HLA-A2^POS^ PD-L1^NEG^ 624-MEL or MCF7 cells are pulsed with a human gp100_280–288_ peptide. These cells are co-cultured with PD-1^POS^ 2D3 cells expressing a TCRαβ recognizing gp100_280–288_ in the context of HLA-A2 either or not in combination with 360 nM of sdAbs or mAbs. When the gp100_280–288_ peptide is recognized by the TCR on the 2D3 cells, this will lead to eGFP upregulation by the 2D3 cells (**right**). (**E**,**F**) Both avelumab and sdAb K2 revert the suppressive effect of PD-L1 expressed on 624-MEL (**E**) or MCF7 (**F**) cells on the activation of PD-1^POS^ 2D3 reporter cells. The graphs depict the reduction in eGFP expression by PD-1^POS^ 2D3 cells when co-cultured with PD-L1^POS^ cancer cells compared to their co-culture with PD-L1^NEG^ cancer cells (mean ± SEM, *n* = 3). No treatment, a control sdAb or an isotype control mAb served as negative controls. Percentage eGFP^POS^ CD8^POS^ PD-1^POS^ 2D3 cells was evaluated using flow cytometry. * *p* < 0.05, ** *p* < 0.01, *** *p* < 0.001.

**Figure 7 cancers-11-00872-f007:**
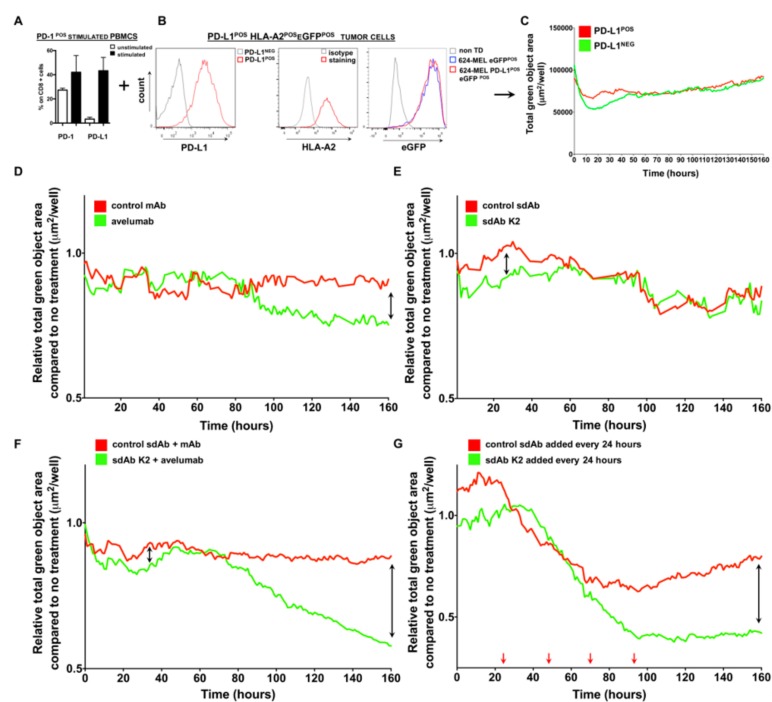
sdAb K2 and avelumab show in vitro therapeutic efficacy, albeit with different kinetics. (**A**–**C**) Schematic representation of the experimental setup. (**A**) Bar graph showing the percentage of PD-1 or PD-L1 expression on CD8^POS^ T cells present within unstimulated PBMCs (white bars) or PBMCs stimulated with anti-CD3 mAb and IL-2 (black bars). The mean ± SEM are shown (*n* = 3). (**B**) Representative histograms showing expression of PD-L1, HLA-A2 and eGFP by engineered 624-MEL cells (*n* = 3). (**C**) Activated PBMCs were added to HLA-A2^POS^, eGFP^POS^ and PD-L1^POS^ 624-MEL cells or to HLA-A2^POS^, eGFP^POS^ and PD-L1^NEG^ 624-MEL cells that were grown in 3D spheroids. The total green object area (μm2/well), representing viable tumor cells, was measured every hour for seven constitutive days using the IncuCyte Zoom^®^ live cell imaging system and is shown in the graph (*n* = 3). (**D**–**G**) Activated PBMCs were added to HLA-A2^POS^, eGFP^POS^ and PD-L1^POS^ 624-MEL cells that were grown in 3D spheroids in the absence or presence of PD-L1 blocking agents. The total green object area (μm2/well), representing viable tumor cells, was measured every hour for seven constitutive days using the IncuCyte Zoom^®^ live cell imaging system and is shown in the graphs. (**D**) Tumor cell killing in the presence of 3.6 μM avelumab (green curve) or 3.6 μM control mAb (red curve). (**E**) Tumor cell killing in the presence of 3.6 μM sdAb K2 (green curve) or 3.6 μM control sdAb (red curve). (**F**) Tumor cell killing in the presence of 3.6 μM sdAb K2 and 3.6 μM avelumab (green curve) or 3.6 μM control sdAb and 3.6 μM control mAb. (**G**) Tumor cell killing when 3.6 μM sdAb K2 (green curve) or 3.6 μM control sdAb (red curve) is added every 24 hours (indicated with red arrows).

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
