# Peer review of "Evaluating a Single Domain Antibody Targeting Human PD-L1 as a Nuclear Imaging and Therapeutic Agent"

_cancers, 2019, doi:10.3390/cancers11060872_

Round 1
Reviewer 1 Report
The authors have characterized in detail a novel single domain antibody termed K2 recognizing PD-L1 molecule, It appears that this antibody can well identify PD-L1+ cells in vivo after radiolabelling with Technetium-99m. In addition, it recognizes a functional epitopes of PD-L1 and it impairs the negative signal delivered in T cells uopn PD-1/PD-L1 interaction.
This work is well presented and of strong interest for the possible applications of this reagent in humans.
Author Response
We sincerely thank the reviewer for the positive evaluation of our work. We performed a spelling
check to ensure that the revised text is without typographical errors.
Reviewer 2 Report
The authors made a high-affinity anti-human PD-L1 specific single domain antibody (K2) and demonstrated both diagnostic and therapeutic utility for PD-L1 inhibitor in the mouse xenograft tumor models. Recent technologies for cancer immunotherapy are focused on. Therefore this study has a potential to lead to re-break through of immunotherapy. However there are some problems to be solved in this study.
<Major>
# This study lacked human immune system in this mouse xenograft model.
If possible, the authors had better make fully-humanoid mice by adding human immune cells (PBMC) to these mouse model, because immune cells expressed PD-L1 also became a target of K2.
# The authors should describe safety data of this K2 in tumor bearing mouse. Pathological evaluation of major organs such as lung, liver, brain or kidney will enhance the significance of this study.
Author Response
To: The Editor of Cancers,
Concerning: Revision of the manuscript entitled “Evaluating a single domain antibody targeting human PD-L1 as a nuclear imaging and therapeutic agent”, reference 515662
Dear Editor,
We would like to thank the editor for considering our manuscript for publication in Cancers. We furthermore thank all reviewers for their valuable remarks. We believe that by taking the reviewers’ comments into account, we were able to strengthen the manuscript. Therefore we hope that the manuscript in its present form fits all requirements for publication in Cancers.
Enclosed, we provide our feedback to the reviewers’ questions in a point-by-point reply.
Sincerely,
On behalf of the authors
Katrijn Broos

Reviewer 3 Report
In this interesting manuscript the authors report the development of a new anti human PD-L1 nanobody. The diagnostic use of this antibody, to stratify patients for PD-L1 expression in their tumors, appears interesting and well described. The use for therapeutic purposes, however, is not completely convincing. Nevertheless, the diagnostic applicability is of great importance too.
A major amendment required relates to the description of data in Figures 6 and 7, that is hard to follow both in the text and Figure legends. More details are required to improve clarity of description of experiments and their interpretations.
Author Response
Responses
We sincerely thank the reviewer for the evaluation of our work and for the valuable remarks.
Below we provide a point-by-point answer to the reviewer’s comments.
Response 1:
To comply with the reviewers’ remark that moderate changes to the text are required, we
performed a thorough spelling check to ensure that the revised text is without typographical
errors.
Response 2:
In the revised manuscript we have changed the description of the data of Figure 6 and 7 both in
the text and figure legends. For the reviewers’ convenience, we have highlighted these sections
in yellow.
Round 2
Reviewer 2 Report
The authors demonstrated the basic mechanism of new type of anti-cancer agent, single domain antibody for PD-L1 in detail. The study has a great significance as a new agent for cancer treatment strategy.